# Improvement of Water and Nitrogen Use Efficiencies by Alternative Cropping Systems Based on a Model Approach

**DOI:** 10.3390/plants12030597

**Published:** 2023-01-29

**Authors:** Le Han, Yunrui Li, Yonghao Hou, Hao Liang, Puyu Feng, Kelin Hu

**Affiliations:** 1College of Land Science and Technology, China Agricultural University, Key Laboratory of Arable Land Conservation (North China), Ministry of Agriculture, Beijing 100193, China; 2College of Agricultural Science and Engineering, Hohai University, Nanjing 210098, China

**Keywords:** crop rotation, soil–crop system model, water balance, water and N use efficiencies

## Abstract

The conventional double cropping system of winter wheat and summer maize (WW-SUM) in the North China Plain (NCP) consumes a large amount of water and chemical fertilizer, threatening the sustainable development of agriculture in this region. This study was based on a three-year field experiment of different cropping systems (2H1Y—two harvests in one year; 3H2Y—three harvests in two years; and 1H1Y—one harvest in one year). The 2H1Y system had three irrigation–fertilization practices (FP—farmer’s practice; RI—reduced input; and WQ—Wuqiao pattern in Wuqiao County, Hebei Province). A soil–crop system model (WHCNS—soil water heat carbon nitrogen simulator) was used to quantify the effects of different cropping systems on water and nitrogen use efficiencies (WUE and NUE, respectively), and to explore the trade-offs between crop yields and environmental impacts. The results showed that annual yield, water consumption, and the WUE of 2H1Y were higher than those of the 3H2Y and 1H1Y systems. However, local precipitation during the period of crop growth could only meet 65%, 76%, and 91% of total water consumption for the 2H1Y, 3H2Y and 1H1Y systems, respectively. Nearly 65% of irrigation water (groundwater) was used in the period of wheat growth that contributed to almost 40% of the annual yield. Among the three patterns of the 2H1Y system, the order of the WUE was 2H1Y_RI > 2H1Y_WQ > 2H1Y_FP. Compared to 2H1Y_FP, the total fertilizer N application rates in 2H1Y_WQ, 2H1Y_RI, and 3H2Y were reduced by 25%, 65%, and 74%, respectively. The 3H2Y system had the highest NUE of 34.3 kg kg^−1^, 54% greater than the 2H1Y_FP system (22.2 kg kg^−1^). Moreover, the 3H2Y system obviously reduced nitrate leaching and gaseous N loss when compared with the other two systems. The order of total N loss of different cropping systems was 2H1Y (261 kg N ha^−1^) > 1H1Y (78 kg N ha^−1^) > 3H2Y (70 kg N ha^−1^). Considering the agronomic and environmental effects as well as economic benefits, the 3H2Y cropping system with optimal irrigation and fertilization would be a promising cropping system in the NCP that could achieve the balance between crop yield and the sustainable use of groundwater and N fertilizer.

## 1. Introduction

The North China Plain (NCP), an area of approximately 32 million ha and 17.98 million ha of agricultural land, is one of the most important grain production regions and one of the main irrigated regions in China. In order to make full use of light and heat resources and achieve a high grain production, the intensive winter wheat–summer maize (WW-SUM) double cropping system with two harvests in one year (2H1Y) has been adopted widely in this region [1,2]. Although this cropping system has significantly enhanced grain production, it has also brought severe issues such as groundwater depletion and environmental pollution [3,4]. The distribution of precipitation in China has been uneven in the last 50 years. As a result, precipitation in the western basins of China has increased the most, with a maximum amplitude of 10–15% per decade, while precipitation has decreased in North China and the south of Northeast China [5]. As a result, natural precipitation does not provide enough water for crop growth in the NCP, especially for the growth of winter wheat. Only 20–30% of the annual rainfall occurs during the winter wheat growing season, and an estimated >400 mm of supplementary irrigation is required to achieve a high yield [6,7,8]. Due to the problem of serious water shortage [9], groundwater is almost the only source of irrigation water in the NCP [10,11]. In the past, the irrigation demand for agricultural production accounted for approximately 70% of the total groundwater consumption [12,13]. To ensure crop production, farmers normally pumped large amounts of groundwater for flood irrigation, which has led to a decline of between 0.5 and 1 m yr^−1^ in the groundwater table since the 1960s [14,15]. Moreover, climate change in China was consistent with global climate change. Several climate change scenarios predicted a decrease in summer precipitation [16], an increase in soil drought [17], and a decrease in river flow [18]. This indicates that the water scarcity problem in the NCP will persist and become more severe in the future.

In addition, water contamination and air pollution due to the excessive application of nitrogen (N) fertilizer are also serious problems. The N application is crucial for intensive grain production. However, fertilizer N recovery by crops is generally poor, so an unreasonable utilization of N fertilizer in intensive cropping systems will result in large amounts of N loss for the environment, contributing to unintended environmental problems [19,20]. To achieve the maximum yield, approximately 500 to 600 kg N ha^−1^ yr^−1^ of fertilizer is normally applied in the double cropping system in the NCP, which far exceeds the basic crop N requirements of 200 to 300 kg N ha^−1^ yr^−1^ [21,22]. Excessive N fertilizer input aggravates N losses, which are mostly linked to gaseous N emissions (N_2_O and NH_3_) and nitrate leaching. This leads to a series of environmental problems: for instance, groundwater nitrate contamination, surface water eutrophication, and air pollution [2,23,24]. Therefore, it is urgent to pay attention to the sustainable groundwater and N fertilizer use to ensure sustainable crop production in the NCP [3,14].

Considering that the WW-SUM double cropping system relies heavily on groundwater resources and chemical fertilizers, many studies have been conducted to optimize irrigation and N use for the conventional cropping system [1,25,26], or to design alternative cropping systems that enhance water and N use efficiencies (WUE and NUE, respectively) and reduce negative environmental impacts while maintaining grain yields in the NCP [2,20,27]. Xiao et al. [26] reported that in comparison with the conventional WW-SUM cropping system, with reductions of 43% and 28% in N fertilizer and groundwater inputs, respectively, the recommended farming management for the double cropping system increased the NUE and irrigation water use efficiency by 79.3% and 61.7%, respectively, while maintaining the mean annual grain yield of 16.5 Mg ha^−1^ in Huantai County, NCP. By conducting a field experiment, Liu [28] studied three cropping systems: conventional double cropping system (2H1Y), winter wheat–summer maize–spring maize (WW-SUM-SPM) with three harvests in two years (3H2Y), and spring maize with one harvest in one year (1H1Y). The results showed that the grain production, water consumption, WUE, and economic benefits of the 2H1Y system were significantly higher than those of the other cropping systems. However, precipitation during the growth period of this pattern only met 72.9% of the total water demand. Additionally, the NUE was 78 kg kg^−1^, significantly lower than the other cropping systems. Sun et al. [29] found that the net consumption of groundwater required for the WW-SUM pattern surpassed 300 mm yr^−1^, and the SPM pattern showed a higher WUE and potential for the sustainable use of groundwater in the long term. The water usage of the WW-SUM-SPM pattern was between the other two patterns, and the grain yield in the WW-SUM pattern was higher than that of the other two patterns (SPM and WW-SUM-SPM). Wang et al. [2] indicated that the order of the annual average yields of the three different cropping systems in the NCP was WW-SUM > WW-SUM-SPM > SPM, but the N leaching of WW-SUM was much larger than the other two systems. The WUE and NUE of the SPM were the highest, followed by WW-SUM-SPM and WW-SUM, respectively. Therefore, the water and fertilizer management sustainability of the “WW-SUM” rotation pattern needs further discussion and optimization. Additionally, in order to reach a balance between grain production and groundwater consumption in the NCP, it is necessary to explore alternative cropping systems and develop best field management practices [14,29,30].

Moreover, to explore the optimal cropping system, which should have a high yield and be environmentally friendly, there is still a lack of quantitative analysis tools for soil water and N fluxes and their efficiencies for different cropping systems. The dynamic field observation of evapotranspiration, water drainage, N losses, and crop biological indicators are time-consuming and costly. To overcome these shortcomings, the soil–crop system models were used to simulate crop growth and N losses in croplands under different water and N management and environmental conditions. Recently, an integrated soil–crop model (WHCNS, soil water heat carbon nitrogen simulator) was developed based on the Chinese climate, soil types, and field management to optimize water and N management [31]. This model has been successfully applied to simulate water consumption, N loss, WUE and NUE in different regions of China [22,32,33,34,35].

Therefore, based on a three-year field experiment involving the three cropping systems (1H1Y, 2H1Y, and 3H2Y) conducted in Wuqiao County, the objectives of this study were to: (1) calibrate and test the feasibility of the WHCNS model in the NCP under different cropping systems, (2) analyze the effects of different cropping systems on water consumption, N fates, WUE, and NUE using the WHCNS model, and (3) compare agronomic, environmental, and economic effects under different cropping systems and identify a sustainable cropping system.

## 2. Materials and Methods

### 2.1. Study Area

The study was conducted from October 2004 to October 2007 at the China Agricultural University experimental station in Wuqiao County (37°29′–37°47′ N; 116°19′–116°42′ E) in Hebei Province, China. The study area has a temperate, sub-humid–arid monsoon climate. The altitude is 14–22.6 m above sea level. The annual mean air temperature was 12.6 °C, and the annual cumulative temperature (≧0 °C) was approximately 4863 °C. The average annual rainfall over the last 20 years was 562 mm, with a sharp yearly fluctuation and an erratic seasonal distribution. Generally, 60–70% of the yearly precipitation occurs from June to August. The soil at the site is classified as Calcaric Fluvisol with a silt loam texture. Irrigated cropland accounts for approximately 70% of the total land in Wuqiao County. The WW–SUM double cropping system is adopted widely in this area. Groundwater is the major water resource for irrigation because rivers are inaccessible. The groundwater depth in the study area is approximately 17 m.

### 2.2. Field Experiment

The experimental design is illustrated in Figure 1. There were three different cropping systems: 2H1Y (WW-SUM, two harvests in one year), 3H2Y (WW-SUM-SPM, three harvests in two years), and 1H1Y (SPM, one harvest in one year). The 2H1Y system had three water and N management practices, namely, 2H1Y_FP (farmer’s practice), 2H1Y_RI (reduced input) and 2H1Y_WQ (Wuqiao pattern). Winter wheat was planted in the middle of October and harvested in the middle of June. The sowing and harvesting dates for summer maize were in the middle of June and early October, respectively. When compared with summer maize, the spring maize had a long crop growth period and was planted and harvested in late April and early September, respectively. For winter wheat, 75 mm of irrigation water was applied on winter wheat at the key crop-growth stages of sowing, seedling, jointing, and grain filling for the 2H1Y_FP management process (Figure 1). The 2H1Y_WQ treatment reduced the irrigation event at grain filling when compared to 2H1Y_FP. The 2H1Y_RI treatment was the reduced water input treatment, and the average water content of the soil in the root zone (0–90 cm) was kept between 50% and 80% of the field capacity based on the measured water content in the soil at the key crop-growth stages of sowing, seedling, and jointing. Water management for the summer maize was similar among treatments and was based on the local farmer’s practice. The optimal amount of N fertilizer application was determined based on the real-time monitoring of the soil mineral N content in the 0–90 cm soil profile. The crop N demand was determined by the crop target yield, and the amount of fertilizer application was the difference between the crop N demand and the soil N supply [19]. Water and N inputs for the 3H2Y and 1H1Y systems were also optimized (Figure 1). Each experiment plot was 10 m × 5 m in size, and experiments were replicated four times in randomized complete block designs. The amount and timing of fertilization and irrigation, the crop planting and harvesting dates, and other field management can be found elsewhere [28,32].

A 1.5 m soil profile pit was excavated, and samples from each soil textural layer were collected before the experiment which began in 2004. The soil bulk density, texture, saturated water content, field capacity, wilting point, and saturated hydraulic conductivity were measured for each layer (Table 1).

In each plot, the soil volumetric water content was measured every 10 days using time domain reflectometry (TDR) (MP-917, ESI, Canada) at 15 cm intervals through 120 cm of the soil profile. Soil samples from the depths of 0–30, 30–60, and 60–90 cm were collected three times: before sowing, at the middle stage, and after harvesting. Each fresh soil sample was extracted with 2 mol L^−1^ of KCl to determine the concentration of NH_4_^+^-N and NO_3_^−^-N using a continuous flow analyzer (TRAACS 2000, Bran and Luebbe, Norderstedt, Germany). The crop leaf area index, dry weights, and N contents of all parts were determined at each key crop development stage. Meteorological data were acquired from a local weather station, including the daily average temperature, minimum and maximum temperatures, relative humidity, average wind speed (2 m), solar radiation, and precipitation. More measurement details were described by Liang et al. [32].

### 2.3. Model Description

Water, N fluxes, and crop growth under various cropping systems were simulated by the WHCNS model [31]. The model was developed as a water and N management tool for intensive cropping systems and agricultural management practices in the NCP, including five main modules: soil water, soil heat, soil C, soil N, and crop growth. The Penman–Monteith method from the FAO was applied to calculate the reference crop evapotranspiration [36]. The method for simulating soil water movement and heat transfer was directly introduced from the HYDRUS-1D [37] and RZWQM [38] models. The PS123 model from the Netherlands was used to simulate crop growth [39]. The simulation of C and N cycles was conducted by the Daisy model from Denmark [40]. The model runs on a daily time step and is driven by meteorological and crop biological variables as well as agricultural management practices. The soil water infiltration and redistribution processes were described by Green–Ampt and Richard’s equations, respectively. Soil N transport simulation was based on the modified convection–dispersion equation. The source-sink term of N transformation and transport includes the mineralization of soil organic N, immobilization in biomass, urea hydrolysis, ammonia volatilization, nitrification, denitrification, and crop uptake. The compensatory absorption mechanism was introduced in the crop water and N uptake [37]. The organic matter pools were divided into three active and three stable C pools. The improved version of the PS123 model was applied to simulate the crop development stage, dry matter production and allocation, and crop yield. The crop yield under water and N stress was calculated based on the simulation of potential and actual crop water and N uptake. A detailed model description is available in the literature [31].

### 2.4. Model Parameters

#### 2.4.1. Soil Hydraulic and Solute Transport Parameters

The soil–water retention characteristic curve and unsaturated hydraulic conductivity are described using the Brooks–Corey [41] and Mualem [42] equations, respectively. The soil saturated water content (*θ_s_*), field capacity (*θ_fc_*), wilting water content (*θ_wp_*), and residual water content (*θ_r_*) were experimentally measured and are listed in Table 1 for their subsequent use in the calculation of hydraulic parameters by using the methods by Rawls et al. [43] and Ma et al. [44].
(1)$$\lambda =\frac{\ln[(\theta_{fc}-\theta_{r})/(\theta_{wp}-\theta_{r})]}{\ln(15000/333)}$$
(2)$$h_{b}=\exp[\frac{\ln(\theta_{fc}-\theta_{r})-\ln(\theta_{s}-\theta_{r})+\lambda \ln(333)}{\lambda }]$$
where *λ* and *h_b_* are the parameters of Brooks–Corey equation, representing the shape coefficient and air-entry value, respectively. The hydraulic dispersion coefficient is given by Equation (3).
(3)$$D_{sh}(v,\theta )=D_{L}\frac{\left|q\right|}{\theta }+D_{0}\tau$$
where *D*_0_ is the molecular diffusion coefficient in free water (cm^2^ d^−1^), |*q*| is the absolute value of the Darcian fluid flux density (cm d^−1^), and *D_L_* is the longitudinal dispersity (cm). The value of *D_L_* is set at 3 cm, and the nitrate and ammonia values of *D*_0_ are 2.4 cm^2^ d^−1^ and 1.2 cm^2^ d^−1^, respectively, based on the literature [31]. Finally, *τ* is a tortuosity factor in the liquid phase (-); its value is provided by Millington and Quirk [45], τ = *θ*^7/3^/*θ_s_*^2^.

#### 2.4.2. Carbon and Nitrogen Transformations

The scope of microbial activity in the soil (soil depth) can be set in the WHCNS model, and the soil N transformation processes (mineralization–immobilization, ammonia volatilization, nitrification, and denitrification) are considered within it. In this case, the soil microbial activity ranged from 0 to 50 cm. The parameters of N transformation referred to the default values of the Daisy model and were calibrated according to the actual situation (Table 2).

The dynamic simulation of organic matter turnover for the long term was not the focus of this study, so the parameters of organic matter decomposition or decay rate originated from earlier studies [40,46], which are shown in Table 2. The initial C/N ratio of residue and various organic matter distribution coefficients were provided by Jensen [47] and are listed in Table 2.

#### 2.4.3. Crop Parameters

The basic crop parameters for crop modeling were obtained from the literature of Driessen and Konijn [39] and Liang et al. [31]. The partition coefficients and maximum assimilation rate (AMAX) were calibrated to match the measured value of dry mass. The crop parameters are shown in Table 3. 

### 2.5. Model Performance Criteria

In order to evaluate the model performance, we applied the following three statistical indices to evaluate the agreement between the simulated and measured values. 

(1) The normalized root mean square error (*nRMSE*):(4)$$nRMSE=\frac{1}{\bar{O}}\cdot \sqrt{\frac{\sum_{i=1}^{n}{\left(S_{i}-O_{i}\right)}^{2}}{n}}$$

(2) Index of agreement (*IA*) [36]
(5)$$IA=1-\frac{\sum_{i=1}^{n}{\left(S_{i}-O_{i}\right)}^{2}}{\sum_{i=1}^{n}{\left(\left|S_{i}-\bar{O}\right|+\left|O_{i}-\bar{O}\right|\right)}^{2}}$$

(3) Nash and Sutcliffe efficiency (*NSE*) [37]:(6)$$NSE=1-\frac{\sum_{i=1}^{n}{\left(S_{i}-O_{i}\right)}^{2}}{\sum_{i=1}^{n}{\left(O_{i}-\bar{O}\right)}^{2}}$$
where *n* is the number of samples, *S_i_* and *O_i_* are the simulated and measured values, respectively, $\bar{O}$ is the mean of the measured data, and *nRMSE* represents the percentage of the average deviation from the measured mean. *nRMSE* < 15%, *nRMSE* = 15–30%, and *nRMSE* > 30% are considered “good,” “moderate,” and “poor” agreements, respectively. The *IA* is an additional method for the evaluation of modeling performance results in a range between 0 and 1. The closer *IA* is to 1, the better the model performs. The *NSE* ranges between −∞ and 1. An efficiency of 1 corresponds to a perfect fit between the simulated values and the observed data, whereas an efficiency less than zero (−∞ < *E* < 0) shows that the model gave poor performance.

### 2.6. Osculating Value Method of Optimizing Water and Nitrogen Management

The osculating value method has been widely used in the assessment of agricultural economic projects and groundwater quality [48,49], and was selected to evaluate and optimize the management of water and fertilizer in this study. For a water and fertilizer management study, there are *n* scenarios (*Q*_1_*, Q*_2_*, … Q_n_*) and *m* assessment indices (*A*_1_*, A*_2_*, … A_m_*). Firstly, we established an initial decision matrix. We then normalized each element as a dimensionless decision matrix (7) that can be written as:(7)$$C=[\begin{array}{cccc} C_{11} & C_{12} & \cdots & C_{1m}\\ C_{21} & C_{22} & \cdots & C_{2m}\\ \cdots & \cdots & \cdots & \cdots \\ C_{n1} & C_{n2} & \cdots & C_{nm} \end{array}]$$
where *C_ij_* is the value of normalized assessment indices (*i* = 1, 2, …, *n*; *j* = 1, 2,… *m*).

Secondly, a target value can be calculated by:(8)$$r_{ij}=\pm C_{ij}/\sqrt{\sum_{i=1}^{m}C_{ij}^{2}}$$
where the symbol “+” identifies the forward factor and “–” identifies the backward factor. Therefore, the dimensionless decision matrix (6) was established:(9)$$R={\left(r_{ij}\right)}_{m\times n}$$

Thus, a theoretical optimal value set, *Q_G_*, and the worst value set, *Q_B_*, were then shown by Lou [49]:(10)$$Q_{G}={\left(r_{ij}\right)}_{G}=\left(\min\{r_{i1}\},\min\{r_{i2}\},\cdots ,\min\{r_{in}\}\right)$$
(11)$$Q_{B}={\left(r_{ij}\right)}_{B}=\left(\max\{r_{i1}\},\max\{r_{i2}\},\cdots ,\max\{r_{in}\}\right)$$

Thirdly, the Euclidean distances, the distances between the assessment indices and the theoretical optimal value set, *d_i_*_-*G*_, and the worst value set, *d_i_*_-*B*_, were determined by the Equations (12) and (13) and are written as:(12)$$d_{i-G}={\left\{\sum_{j=1}^{n}\omega_{j}•{\left[r_{ij}-{\left(r_{ij}\right)}_{G}\right]}^{2}\right\}}^{1/2}$$
(13)$$d_{i-B}={\left\{\sum_{j=1}^{n}\omega_{j}•{\left[r_{ij}-{\left(r_{ij}\right)}_{B}\right]}^{2}\right\}}^{1/2}\;$$
where *ω_j_* is the weight of the assessment index *j*, $\sum_{j=1}^{n}\omega_{j}=1$.

Finally, the osculating value, *E_i_*, was calculated. Depending on the extreme value, the osculating value is the optimal osculating value *E_i_*_-*G*_ (14) and the worst osculating value *E_i_*_-*B*_ (15)_._:(14)$$E_{i-G}=\frac{d_{i-G}}{\underset{1\leq i\leq m}{\min}\{d_{i-G}\}}-\frac{d_{i-B}}{\underset{1\leq i\leq m}{\max\{d_{i-B}\}}}\;$$
(15)$$E_{i-B}=\frac{d_{i-B}}{\underset{1\leq i\leq m}{\min}\{d_{i-B}\}}-\frac{d_{i-G}}{\underset{1\leq i\leq m}{\max\{d_{i-G}\}}}\;$$

The scenario with the lowest *E_i_*_-*G*_ and the highest *E_i_*_-*B*_ is the best.

Agronomy effects (AFs), environmental effects (EFs), and the value-to-cost ratio (VCR) were taken into account in calculating the osculating values. Referring to Sun et al. [50] and Xu et al. [51], the grain yield, WUE, and NUE were selected as the evaluation indices to calculate the AF. The weights of the yield, WUE, and NUE were set as +0.6, +0.1, and +0.3, respectively. The three indices were normalized with a range of 0 to 1. The AF was calculated by summing the product of the normalized indices and their corresponding weights. Gaseous N loss and nitrate leaching were selected as the evaluation indices for calculating the EF. The weight of gaseous N loss and nitrate leaching were set as +0.5 and +0.5. After the two indices were normalized, the EF was calculated using a method similar to the calculation of the AF. The *VCR* was calculated by Equation (16):(16)$$VCR=\frac{\left(Y\times YP\right)}{\left(W\times WP+F\times FP\right)}$$
where *W* is the irrigation amount (mm), *WP* is the water price (CNY m^−3^), *F* is the fertilizer application rate (kg N ha^−1^), *FP* is the fertilizer price (CNY kg^−1^), *Y* is the grain yield (kg ha^−1^), and *YP* is the grain price (CNY kg^−1^). In this study, the original prices of water, fertilizer, winter wheat, and summer maize were 0.25 CNY m^−3^, 4.5, 2.4, and 2.0 CNY kg^−1^, respectively.

In this study, 25 scenarios (*n* = 25 in Equation (7)) with different water prices were set as follows: 5 systems (2H1Y_FP, 2H1Y_WQ, 2H1Y_RI, 3H2Y, and 1H1Y) × 5 kinds of water prices (0.25, 0.5, 1, 1.5, and 2 CNY m^−3^). There were 3 assessment indices in this study (*m* = 3 in Equation (7)): AF, EF, and VCR. The weights of AF, EF, and VCR were set to +0.7, −0.2, and +0.5, respectively [50].

### 2.7. Statistical Analysis

The data were recorded and processed using Excel. Comparisons of the groups were performed using the one-way analysis of variance (ANOVA) with a Duncan multiple range test (*p* < 0.05). The correlation coefficients were calculated with the SPSS 26 software (SPSS Inc., Chicago, IL, USA), using Pearson’s correlation and a two-tailed t test (*p* < 0.05 and *p* < 0.01). Origin 2021 was used for plotting data.

## 3. Results

### 3.1. Model Calibration and Validation

The model ran continuously from 16 October 2004 to 2 October 2007. Data from the 3H2Y system, including soil water content, soil nitrate concentration, crop LAI (leaf area index), crop dry mass, and crop N uptake and yield were used to calibrate the parameters of N transformation and the crop parameters of winter wheat, summer maize and spring maize. Finally, data from treatments 2H1Y_FP, 2H1Y_WQ, 2H1Y_RI, and 1H1Y were used to validate the model. Figure 2 and Figure 3 illustrate the measured and simulated soil water content and soil nitrate concentration for the 2H1Y_RI system at each soil depth. Only the results of the 2H1Y_RI system are presented; the simulation results for the other treatments are provided in the Appendix A). In general, the simulated data agreed well with the measured data. Table 4 shows that there were significant correlations between the measured and simulated volumetric water content, nitrate concentration, crop dry mass, crop LAI, and crop N uptake for all five systems, and their coefficients of determination were 0.519, 0.317, 0.916, 0.522, and 0.828, respectively.

The *nRMSE*, *NSE*, and *IA* between the simulated and measured soil water content were 20.3%, 0.423 and 0.847, respectively (Table 4). Van Liew and Garbrecht [52] analyzed many soil water process models, considering that the soil water model performed well when the *NSE* > 0.36 and the *IA* > 0.7. For the simulated soil nitrate concentration, the *nRMSE*, *NSE*, and *IA* were 37.5%, 0.025, and 0.708, respectively (Table 4). In fact, the model performance criteria are different for different researchers. Yang et al. [53] recommended values of IA ≥ 0.75 and NSE ≥ 0 as the minimum threshold values for crop growth evaluation, and recommended values of IA ≥ 0.60 and NSE ≥ −1.0 as the minimum threshold values for soil output evaluation. Kersebaum [54] compared the performance of thirteen soil–crop models on soil water and N dynamics and found that a negative value of the *NSE* can be accepted due to the complexity of the N transformation. Therefore, the simulation results of this study can be considered acceptable.

Regarding the crop growth simulation, the *nRMSE* of the crop dry mass, crop LAI, and crop nitrogen uptake and yield were 19.4%, 29.9%, 21.2% and 12.9%, respectively. The *NSE* values > 0.4 and *IA* values > 0.8 for all four crop items indicated that the model had a relatively good performance for crop growth modeling.

### 3.2. Crop Yield, Water Fluxes, and Water Use Efficiency

Figure 4 illustrates grain yields under different cropping systems. The average annual yields of the 2H1Y system were higher than those of the other two cropping systems. The average annual yield of the 1H1Y system was the lowest (7564 kg ha^−1^ yr^−1^), 48% lower than those of the three patterns under the 2H1Y system (14,590 kg ha^−1^ yr^−1^) (Figure 4). The annual yield of the 3H2Y system was 11,737 kg ha^−1^ yr^−1^, approximately 20% lower than those of the three 2H1Y systems. For the 2H1Y systems, the 2H1Y_WQ pattern had the highest annual yield (15,163 kg ha^−1^ yr^−1^), followed by the 2H1Y_FP and 2H1Y_RI patterns, which had average annual yields of 14,400 and 14,260 kg ha^−1^ yr^−1^, respectively (Figure 4). The average yield of the maize season (8660 kg ha^−1^) was 33% higher than that of the wheat season (5810 kg ha^−1^) for the 2H1Y system. The average winter wheat yields of the three 2H1Y systems ranked in the order of 2H1Y_WQ > 2H1Y_FP ≈ 2H1Y_RI, with yields of 6206, 5634, and 5602 kg ha^−1^ per season, respectively. The average yields of summer maize for the 2H1Y_WQ, 2H1Y_FP, and 2H1Y_RI patterns were 8818, 8679, and 8484 kg ha^−1^ per season, respectively (Table 5). Compared with the 2H1Y_FP pattern, the 2H1Y_WQ and 2H1Y_RI treatments with reduced water and N inputs did not cause a significant decline in grain yield and even the average annual yield of the 2H1Y_WQ system increased by 5%.

The water fluxes and WUE under different cropping systems simulated by the WHCNS model are summarized in Table 5. The main water inputs were precipitation and irrigation. The total amount of precipitation in the crop growth period for the 1H1Y system was a minimum of 1105 mm, and the amounts for the 3H2Y and 2H1Y systems were similar: 1316 and 1344 mm, respectively. The total amounts of irrigation for 2H1Y_FP, 2H1Y_RI, 2H1Y_WQ, 3H2Y, and 1H1Y were 1107, 538 882, 482, and 180 mm, respectively. The total irrigation amounts of 2H1Y_WQ, 2H1Y_RI, 3H2Y, and 1H1Y were reduced by 20%, 50%, 56%, and 84% when compared with the 2H1Y_FP pattern. Compared with the 2H1Y_RI pattern, the total irrigation amounts of 3H2Y and 1H1Y were reduced by 56 mm and 358 mm, respectively.

As can be seen from Table 5, evapotranspiration (ET) and drainage (D) were the two main water outputs. ET accounted for 90–95% of the total water consumption for all systems. The order of the total ET of different cropping systems ranked 2H1Y_FP > 2H1Y_WQ > 2H1Y_RI > 3H2Y > 1H1Y. The total water drainage was the lowest for 3H2Y (7 mm) and the largest for 2H1Y_FP (162 mm). The orders of the total ET and water drainage were the same: 2H1Y_FP > 2H1Y_WQ > 2H1Y_RI > 3H2Y > 1H1Y.

A PE value over 1 indicates that the precipitation during crop growth is more than the ET, and that the crop experiences low water stress. Conversely, when the PE value is closer to 0 and no irrigation is applied, the crop development will be under high water stress. The average PE values for all five systems are listed in Table 5. The order of average PE values ranked as: 1H1Y > 3H2Y > 2H1Y_RI > 2H1Y_WQ > 2H1Y_FP. The average PE values of WW, SUM, and SPM were 0.3, 1.04, and 1.13, respectively, which indicated that the WW growth season required a lot of irrigation water input, and nearly 65% of irrigation water (groundwater) was used in the period of wheat growth that contributed to almost 40% of the annual yield. The PE value for the 2H1Y system was the lowest; this was caused by the annual planting of winter wheat, because only 20–30% of the total annual rainfall occurs during the winter wheat growing season in the NCP.

Regarding WUE, the WUE of the 2H1Y_RI pattern was the highest (2.4 kg m^−3^), while the WUE of the 1H1Y system was the lowest (2.1 kg m^−3^). The order of the WUE was: 2H1Y_RI > 2H1Y_WQ > 3H2Y > 2H1Y_FP ≈ 1H1Y. The slightly higher WUE of the 2H1Y system was attributed to the higher total yields of 2H1Y when compared with the other two cropping systems. The WUE of the SUM season was higher than that of the WW season in the 2H1Y cropping system, with average amounts of 3.0 kg m^−3^ and 1.7 kg m^−3^, respectively. The WUEs of three 2HIY systems for the WW and SUM seasons both ranked in the same order: 2H1Y_FP < 2H1Y_WQ < 2H1Y_RI.

Regarding water balance, the order was: 2H1Y_FP > 2H1Y_WQ > 1H1Y ≈ 3H2Y > 2H1Y_RI, with values of 185, 95, 65, 62, and −4 mm, respectively. This indicated that the water input for the 2H1Y_FP pattern far exceeded the crop demand, and the water input and crop demand for 2H1Y_RI reached a relatively balance state.

### 3.3. Nitrogen Fluxes and Nitrogen Use Efficiency

The simulation results of the N fluxes and N use efficiency are listed in Table 6. Chemical fertilizer and mineralization were the main nutrient sources in the agricultural production system. The 1H1Y system had an obviously lower average fertilizer application rate (44 kg N ha^−1^ yr^−1^) compared with those of the 3H2Y (145 kg N ha^−1^ yr^−1^) and 2H1Y (an average of 386kg N ha^−1^ yr^−1^ for three 2HIY patterns) systems. Compared with the 2H1Y_FP pattern, the average application rates of fertilizer in 2H1Y_WQ, 2H1Y_RI, 3H2Y and 1H1Y were reduced by 25%, 65%, 74%, and 92%, respectively (Table 6 and Figure 1).

Crop N uptake, gaseous N loss (the sum of denitrification and ammonia volatilization) and nitrate leaching were the main N output items. Crop N uptake took up the largest proportion of N consumption and accounted for 80–93% of the total N consumption for all systems. Denitrification and ammonia volatilization for different cropping systems ranged from 22 to 80 kg N ha^−1^ and from 31 to 127 kg N ha^−1^, respectively, and they are both in the order of: 2H1Y_FP > 2H1Y_WQ > 2H1Y_RI > 3H2Y > 1H1Y.When compared with 2H1Y_FP, the total gaseous N loss of 2H1Y_WQ, 2H1Y_RI, 3H2Y, and 1H1Y was reduced by 16%, 53%, 67%, and 75%, respectively. The 2H1Y_FP pattern had the largest amount of total nitrate leaching of 187 kg N ha^−1^, while 3H2Y had the minimum amount of total nitrate leaching (1 kg N ha^−1^). When compared with the conventional 2H1Y_FP pattern, the total nitrate leaching amounts of 2H1Y_WQ, 1H1Y, 2H1Y_RI, and 3H2Y dropped by 47%, 87%, 89%, and 99%, respectively.

Regarding the NUE, the 3H2Y and 2H1Y_FP systems had the maximum and minimum NUE (34.3 and 23.6 kg kg^−1^), respectively. The NUE of different systems ranked in the order: 3H2Y > 2H1Y_RI > 1H1Y > 2H1Y_WQ > 2H1Y_FP. The average NUEs of WW, SUM, and SPM were 21.8, 26.2, and 43.0 kg kg^−1^, respectively. This indicated that the annual planting of winter wheat was the main reason for the relatively lower NUE for the 2H1Y system.

Soil N balance demonstrates the net change of soil mineral N stock. The order of N balance of different systems was: 2H1Y_FP > 2H1Y_WQ > 2H1Y_RI > 3H2Y > 1H1Y, with values of 262, 140, −149, −171, and −296 kg N ha^−1^, respectively. The N balance for 2H1Y_FP was all positive, indicating the accumulation of soil mineral N in the soil profile. Negative values for 2H1Y_RI, 3H2Y, and 1H1Y demonstrated the net consumption of soil mineral N. Therefore, both the N fertilizer rates for 2H1Y_FP and even the 2H1Y_WQ pattern with the reduced N input far exceeded crop demands.

### 3.4. Determining the Best Cropping Systems

A quantitative analysis of the model outputs (grain yield, water drainage, WUE, nitrate leaching, gaseous N loss, and NUE) can help determine the best cropping system. The evaluation results are shown in Table 7. Under different water price scenarios, the 1H1Y system had the largest VCR, a relatively lower EF, the lowest OOV (the optimal osculating value) and the largest WOV (the worst osculating value). However, it had the lowest AF, with a lower average annual yield compared to the other four systems (Figure 4 and Table 7). Considering the yield is a very important indicator for determining the best cropping system, we set a limitation that the annual yield of reasonable systems should be larger than 11,000 kg ha^−1^ yr^−1^ (more than 70% of the highest annual yield of 15163 kg ha^−1^ yr^−1^). Regarding the remaining scenarios, 2H1Y_FP had the lowest AF (0.86) and the largest EF (1). The average VCR value of 2H1Y_FP was the lowest at 5.64. Moreover, this cropping system had the largest OVV and the lowest WOV. Thus, 2H1Y_FP was the least eco-friendly and the most unsustainable. The 3H2Y system had the largest average VCR of 13.72 and the lowest EF (0.17). The slightly lower total yield of the 3H2Y system led to a relatively lower AF when compared with the 2H1Y_RI and 2H1Y_WQ systems. After a comprehensive evaluation of the AF, EF and VCR, the 3H2Y system demonstrated the lowest OVV and the largest WOV values, indicating that this system was the best cropping system among the five systems. Under the 3H2Y system, the total yield, WUE and NUE were 36,243 kg ha^−1^, 2.2 kg m^−3^ and 34.3 kg kg^−1^, respectively, while total nitrate leaching and gaseous N loss were 1 and 68 kg N ha^−1^, respectively, far lower than those under the 2H1Y system (Table 5,Table 6).

## 4. Discussion

### 4.1. The Effects of Different Cropping Systems on Crop Yield and Water Use Efficiency

Different cropping systems have different production levels, mainly due to the different crop combinations in these systems, which will inevitably lead to different WUEs. In this experiment, the total yield and total water consumption under the 2H1Y systems were higher than those of the 3H2Y and 1H1Y systems. The WUE of the 2H1Y system was slightly higher than those of the other two systems. This result was similar to the study of Sun et al. [27]; under normal irrigation, the total yield of 2H1Y reached 19,732 kg ha^−1^, while the yields of 3H2Y and 1H1Y were 15,472 and 10,737 kg ha^−1^, respectively. Additionally, the WUE of 2H1Y reached 1.33 kg m^−3^, while the WUE amounts of the other systems were around 1 kg m^−3^. Wang et al. [2] reported that the WUE under the 1H1Y system was higher than the WUEs of 3H2Y and 2H1Y because this pattern had a lower ET. In this study, 2H1Y_RI had the highest WUE. This was because the 2H1Y_RI pattern had a relatively lower annual ET with a relatively higher annual crop yield. This result agreed with the study of Li et al. [1] that the WUE of the optimized 2HIY system was approximately 2.2 kg m^−3^, higher than that of the farmer’s practice (1.8 kg m^−3^). The WUE of the three 2H1Y systems was slightly higher than the other two systems. This was mainly due to most precipitation occurring in the summer season (June–August), which is not synchronized with the water requirement of winter wheat growth [32,55,56].

Considering that the groundwater table is deep in the NCP, we defined the PE index (the precipitation in the crop growth period divided by the actual ET) as the contribution capability of natural precipitation to the crop water requirement. The PE value of WW in this study was only 0.3, indicating that large amounts of groundwater were needed to support the growth of winter wheat. Some studies indicated that irrigation was not necessary for maize growth, but around 250 mm yr^−1^ of groundwater was still needed for irrigation in the winter wheat season in the optimized winter wheat–summer maize rotation system due to the large gap between the precipitation and the water demand [14,56]. This also proves that altering the conventional WW-SUM double cropping system to the alternative cropping systems by reducing winter wheat cultivation is feasible. Wheat and maize belong to the C3 plant and the C4 plant, respectively. Due to different morphologies, anatomical structures and CO_2_ assimilation modes, the water consumptions and yield formations between the species are different, resulting in different WUEs [57]. Maize has a better WUE than winter wheat because it mainly relies on precipitation for growth. SPM has a high yield potential due to its long growth period with the available sunlight resources in the NCP [2]. It is reported that the potential yield of SPM can reach up to 14–23 t ha^−1^ in this region [2]. Therefore, taking full advantage of the high light efficiency and high yield potential of C4 crops such as summer maize and spring maize, optimizing and innovating the traditional WW-SUM rotation pattern has become the key to solving the water shortage problem [58]. However, when making regional decisions on cropping systems, indicators such as crop yield cannot be taken as the only consideration, and actual conditions should also be considered. Therefore, the balance between sustainable groundwater use and crop yield should be comprehensively considered to select the most reasonable cropping system for the NCP.

### 4.2. The Effects of Different Cropping Systems on N Fates and N Use Efficiency

Under the wheat–maize cropping system in NCP, excessive irrigation and N fertilizer application were the main reasons for N leaching. Many studies proposed that reducing the amounts of irrigation and N fertilizer application can significantly reduce N leaching [59,60]. In this study, the total irrigation amounts of 3H2Y, 1H1Y and 2H1Y_RI were lower, so the corresponding N leaching losses were lower than those of 2H1Y_FP and 2H1Y_WQ. When compared with the conventional 2H1Y_FP pattern, the total N leaching of the 2H1Y_WQ, 1H1Y, 2H1Y_RI and 3H2Y systems dropped by 47%, 87%, 89% and 99%, respectively. Our results agreed well with previous studies. Some studies showed that nitrate leaching mainly occurred in the summer maize season [2,61] because 70–80% of the yearly precipitation occurs in the summer season. In this study, N leaching mainly occurred in the maize growing season in 2006, which was related to the largest precipitation (516 mm) in the season. Compared with the SUM season of the 2H1Y system, the N leaching during the period of the SPM season under the 3H2Y and 1H1Y systems was obviously lower, indicating that the long growth of SPM could effectively reduce N leaching in the rainy season. These results were consistent with the reported by Wu et al. [62].

Except for N leaching, ammonia volatilization and denitrification are other main pathways for N loss under different cropping systems. Usually, the amounts of denitrification and ammonia volatilization increase with the increase in fertilizer rates [63,64]. In this study, the order of total gaseous N loss under different cropping systems was: 2H1Y_FP > 2H1Y_WQ > 2H1Y_RI > 3H2Y > 1H1Y. Compared with the 2H1Y_FP system, the total gaseous N loss of the 2H1Y_WQ, 2H1Y_RI, 3H2Y and 1H1Y systems was reduced by 16%, 53%, 67% and 75%, respectively, which was consistent with other studies conducted in the NCP [63,64].

N use efficiency was closely related to the crop yield, N uptake, and N loss. The results of this experiment showed that a reasonable regulation of water and N input could reduce N loss while ensuring grain yield and improving N utilization. Usually, the sowing and harvesting dates of summer maize are in mid-June and early October, respectively, whereas spring maize is sown in late April and harvested in early October [65]. Spring maize had a longer growth period and a larger crop N uptake than summer maize in the NCP, so the NUEs of the 1H1Y and 3H2Y systems were relatively higher than that of the 2H1Y system. Wang et al. [66] found that the N application rate of the 3H2Y system was 75 % lower than that of the 2H1Y system, and the fertilizer NUE was increased by 211%. Guo et al. [67] reported that, compared with the 2H1Y system, the 3H2Y or 1H1Y systems significantly increased the fertilizer NUE, which was consistent with our results.

### 4.3. Developing Sustainable Cropping Systems with High Yield, High Water and N Use Efficiencies

The traditional 2H1Y cropping system in NCP guarantees national food security. However, the system consumes a large amount of groundwater (>300 mm yr^−1^) and chemical N fertilizers (>600 kg Nha^−1^ yr^−1^) in the NCP, which seriously hinders the sustainable agricultural development of this region. To address these challenges, developing high-output, low-pollution agriculture with a highly efficient use of water and fertilizer is necessary to maintain sustainable crop production in the NCP [1,68].

One of the solutions is to optimize water and N management for the traditional 2H1Y cropping system [1,22,25]. Li et al. [1] found that optimized irrigation and fertilizer inputs in the 2H1Y system could reduce the annual average N loss of 28.6% and increase the annual grain yield, WUE, and NUE by 27.7%, 25.7% and 22%, respectively, compared with the local farmer’s practice. Fang et al. [69] reported that the 2H1Y system has great potential for reducing N inputs to increase the NUE and to mitigate N leaching into the groundwater and proposed that controlling irrigation and matching N application to crop N demand is the key to reducing nitrate leaching and maintaining crop yield. In this study, the optimized patterns of 2H1Y_WQ and 2H1Y_RI of the 2H1Y system, with reductions of 20–50% of the irrigation amounts and 25–75% of the N fertilizer rates, not only maintained grain yields, but also increased the WUE and NUE by 7–12% and 18–40%, respectively, when compared with the traditional 2H1Y_FP system (Table 5 and Table 6). In addition, gaseous loss and nitrate leaching were reduced by 16–53% and 47–89%, respectively, compared with the 2H1Y_FP pattern (Table 6). Our results were consistent with the above studies.

Alternative cropping systems with reasonable management practices can be another strategy for developing a sustainable production system in the NCP [14,68]. To reduce water and N use while maintaining productivity at acceptable levels, the 3H2Y cropping system and the 1H1Y monoculture system (e.g., continuous spring maize) have been studied as alternatives to examine the possibility of replacing the conventional 2H1Y system of WW-SUM [2,22,70]. In this study, the 1H1Y cropping system had the lowest average annual water and N input (84% and 92% lower than those in 2H1Y_FP, respectively), which contributed to a lower average annual water drainage and N loss when compared with 2H1Y_FP (reduced by 67% and 80%, respectively). Therefore, this system was eco-friendly. However, when compared with the 2H1Y_FP system, the 1H1Y system caused a large decline in grain yield, with a reduction of 47% in the average annual yield. This led to an unobvious increase in the WUE though a 22% increase in the NUE. Some studies found that crop yield in the monoculture system decreased sharply compared with the conventional 2H1Y system in the NCP [11,29]. Wang et al. [2] also indicated that, although water and fertilizer resource consumption and environmental effects were perfect in the monoculture 1H1Y system, developing this cropping system is impossible in the NCP if food security cannot be guaranteed. Some studies have shown that the 3H2Y cropping system is more sustainable and could maintain high target yields while reducing N losses and the decline of the groundwater table [11,27,30]. Similar results were found in this study: when compared with 2H1Y_FP, the 3H2Y cropping system had much lower water and N inputs, but the reduced inputs did not cause a large decline in the average annual yield (reduced by 16%) (Table 5); the WUE of 3H2Y slightly increased, and its NUE rose by 54%. Moreover, the N loss of the 3H2Y system was reduced by 82% when compared with the 2H1Y_FP system. A comprehensive assessment of the agronomic and environmental effects as well as economic benefits indicated that the 3H2Y system has a stable grain yield and WUE, as well as the lowest N loss and the largest NUE among all five cropping systems. Therefore, it could be a promising cropping system and should be recommended for use in the NCP to achieve the balance between crop yield and the sustainable use of groundwater and N fertilizer.

However, considering that the experiment was conducted many years ago, the wheat yield today reaches 8000–9000 kg ha^−1^ and is much higher than that of many years ago due to the different wheat varieties and field management practices. Many studies also found that the higher yield will result in a higher crop N uptake. The recommended system in this study should be tested based on the new wheat varieties and field management technologies.

## 5. Conclusions

The WHCNS model was calibrated and validated under different cropping systems in the NCP. The simulated soil water content, nitrate concentration in the soil profile, and crop growth agreed well with the measured data. We concluded that the WHCNS model can be used as a tool to simulate the water and N dynamics and the crop growth in the study area.

Under the 2H1Y system, the optimized management of 2H1Y_WQ and 2H1Y_RI not only had less water and N input (with water reduced by more than 20–50% and fertilizer reduced by 25–75%), but also maintained or slightly increased yields and the WUE when compared with 2H1Y_FP. The NUE of 2H1Y_RI and 2H1Y_WQ was much higher than that of 2H1Y_FP (increased by 18–40%), and the N loss was much lower than that of 2H1Y_FP. The optimized 2H1Y_RI obviously reduced nitrate leaching, with the lowest total nitrate leaching value of 20 kg N ha^−1^, compared with the nitrate leaching values of 2H1Y_WQ and 2H1Y_FP, which were 100 kg N ha^−1^ and 187 kg N ha^−1^, respectively.

For the different cropping systems, the annual yield, annual water consumption, and annual WUE of the 2H1Y system were higher than those of 3H2Y and 1H1Y, and the average annual yield of 2H1Y, 3H2Y, and 1H1Y were 14590, 11737, and 7564 kg ha^−1^ yr^−1^, respectively. However, the local precipitation during the period of crop growth could only meet 65%, 76% and 91% of the total water consumption for the 2H1Y, 3H2Y, and 1H1Y systems, respectively, and nearly 65% of the irrigation water (groundwater) was used in the period of wheat growth which contributed to almost 40% of the annual yield. The optimized 3H2Y system obviously increased the NUE. Among the five systems, 3H2Y had the largest NUE of 34.3 kg kg^−1^, while the conventional 2H1Y_FP had the lowest NUE of 22.2 kg kg^−1^. Moreover, 3H2Y had the lowest total nitrate leaching (1 kg N ha^−1^) when compared with the 1H1Y (25 kg N ha^−1^) and 2H1Y (average of 102 N kg ha^−1^) systems.

The 3H2Y system had a slightly lower total yield compared with the 2H1Y system, but it had the largest value-to-cost ratio and the lowest negative environmental effects under different water prices. Therefore, this system could be a promising cropping system for use in the NCP.

## Figures and Tables

**Figure 1 plants-12-00597-f001:**
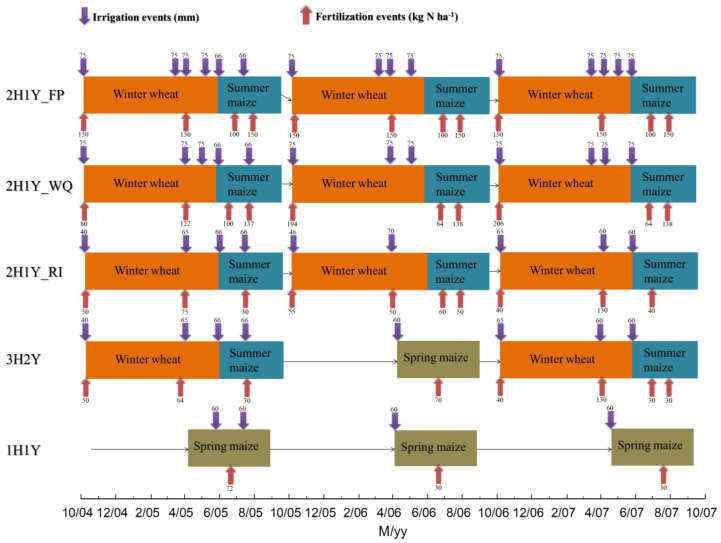
Crop rotations and water and N inputs for five different cropping systems. Note: 2H1Y_FP—two harvests in one year of farmer’s practice; 2H1Y_WQ—two harvests in one year of the Wuqiao pattern; 2H1Y_RI—two harvests in one year of the reduced input; 3H2Y—three harvests in two years; 1H1Y—one harvest in one years; WW—winter wheat; SUM—summer maize; SPM—spring maize; M/yy—Month/Year.

**Figure 2 plants-12-00597-f002:**
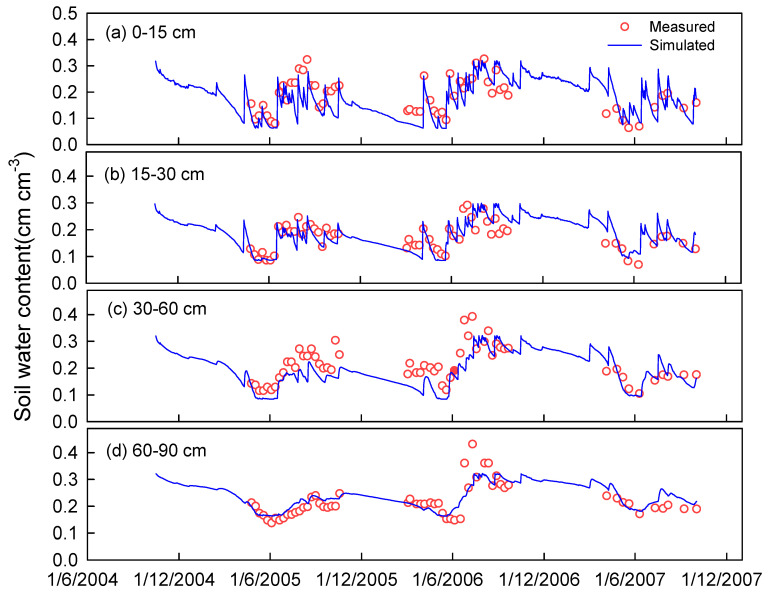
Comparison of simulated and measured soil water content at different depths for 2H1Y_RI.

**Figure 3 plants-12-00597-f003:**
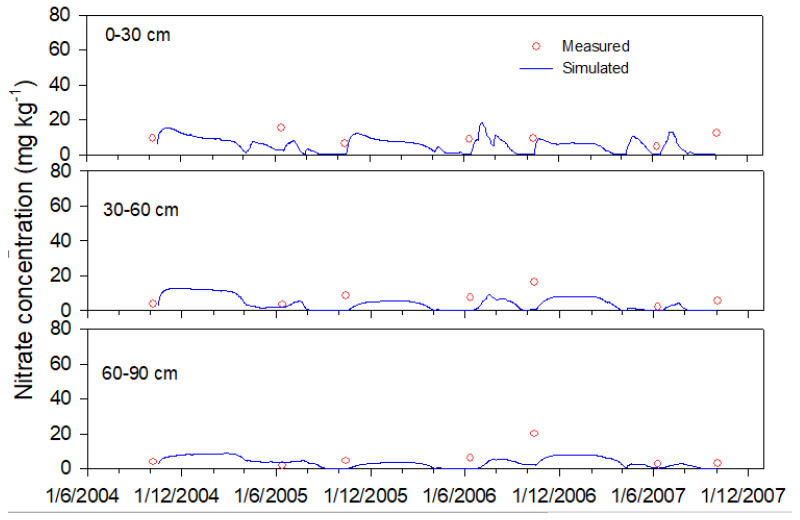
Comparison of simulated and measured soil nitrate N concentration at different depths for 2H1Y_RI.

**Figure 4 plants-12-00597-f004:**
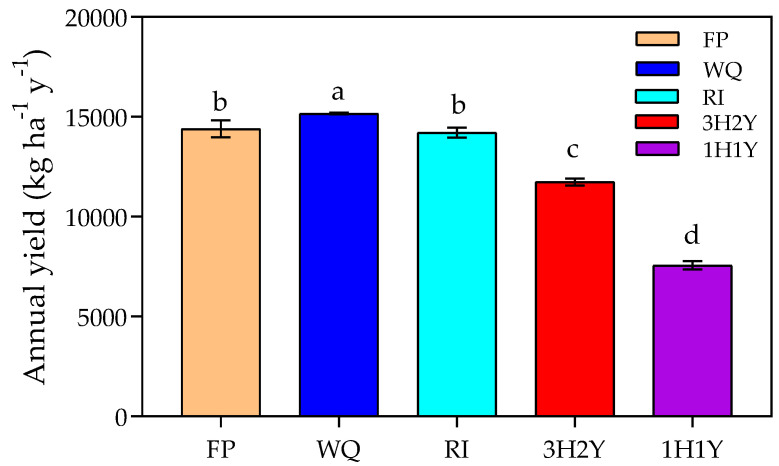
Average annual crop yields under different cropping systems. Error bars represent the standard error. Different lowercase letters indicate significant differences among yields under different treatments (*p* < 0.05).

**Table 1 plants-12-00597-t001:** Soil texture and hydraulic properties of the soil profile in the experiment area.

Soil Layer (cm)	BD (g cm^−3^)	Particle Fraction (%)			Texture (USDA)	*K_s_* (cm d^−1^)	*θ_r_* (cm^3^ cm^−3^)	*θ_s_* (cm^3^ cm^−3^)	*θ_fc_* (cm^3^ cm^−3^)	*θ_wp_* (cm^3^ cm^−3^)
		Sand	Silt	Clay						
0–30	1.54	9	82	9	Silt loam	13	0.054	0.36	0.25	0.084
30–50	1.54	14	76	10	Silt loam	19	0.054	0.33	0.23	0.096
50–75	1.50	5	81	14	Silt loam	20	0.066	0.32	0.21	0.083
75–100	1.44	2	86	12	Silt loam	17	0.067	0.32	0.26	0.154
100–150	1.38	1	76	23	Silt clay loam	3.5	0.079	0.48	0.31	0.228

Note: BD is bulk density, *K_s_* is the soil saturated hydraulic conductivity, *θ_r_* is the residual soil water content, *θ_s_* is the saturated soil water content, *θ_fc_* is the field capacity, and *θ_wp_* is the wilting point.

**Table 2 plants-12-00597-t002:** Soil carbon and nitrogen transformation parameters.

Parameters	Description	Value
	Carbon turnover parameters	
*k^*^_SOM1_*	Decomposition rate of SOM1(d^−1^)	2.70 × 10^−6^
*k^*^_SOM2_*	Decomposition rate of SOM2(d^−1^)	1.40 × 10^−4^
*k^*^_AOM1_*	Decay rate of SMB1(d^−1^)	1.85 × 10^−4^
*k^*^_AOM2_*	Decay rate of SMB2(d^−1^)	1.00 × 10^−2^
*d^*^_BOM1_*	Decomposition rate of plant material—AOM1(d^−1^)	1.20 × 10^−2^
*d^*^_BOM2_*	Decomposition rate of plant material—AOM2(d^−1^)	5.00 × 10^−2^
*m^*^_BOM1_*	Maintenance respiration coefficient of SMB1(d^−1^)	1.80 × 10^−3^
*m^*^_BOM2_*	Maintenance respiration coefficient of SMB2(d^−1^)	1.00 × 10^−2^
*f_SOM2_SOM1_*	Partitioning coefficient, SOM2 to SOM1(-)	0.1
*f_BOM1_SOM2_*	Partitioning coefficient, SMB1 to SOM2(-)	0.6
*f_BOM2_SOM2_*	Partitioning coefficient, SMB2 to SOM2(-)	0.6
*f_AOM1_BOM1_*	Partitioning coefficient, AOM1 to SMB1(-)	0.5
*f_AOM1_BOM2_*	Partitioning coefficient, AOM1 to SMB2(-)	0.5
*[C/N]_SOM1_*	C/N ratio of SOM1 (depending on soil–SOM) (-)	8–14
*[C/N]_SOM2_*	C/N ratio of SOM2 (depending on soil–SOM) (-)	8–14
*[C/N]_BOM_*	C/N ratio of SMB (SMB1 and SMB2) (-)	6.7
*[C/N]_AOM1_*	C/N ratio of AOM1 (depending on input material) (-)	100
*[C/N]_AOM2_*	C/N ratio of AOM2 (depending on input material) (-)	20
	N transformation parameters	
*V_n_*	Maximum nitrification rate (mg N L^−3^ d^−1^)	20
*K_n_*	Half saturation constant for nitrification (mg N L^−3^)	50
*K_d_*	Empirical proportionality factor for denitrification (-)	0.2
*A_d_*	First-order kinetic coefficient for denitrification (d^−1^)	0.03
*K_v_*	First-order kinetic coefficient for ammonia volatilization (d^−1^)	0.1

**Table 3 plants-12-00597-t003:** Crop parameters used in the WHCNS model.

Parameters	Description	Crops		
		WW	SUM	SPM
*Tbase*	Base temperature (°C)	0	8	8
*Tsum*	Accumulated temperature (°C)	1900	1600	2200
*Ke*	Extinction coefficient	0.6	0.6	0.6
*K_ini*	Crop coefficient in initial stage	0.65	0.65	0.65
*K_mid*	Crop coefficient in middle stage	1.25	1.35	1.35
*K_end*	Crop coefficient in end stage	0.6	0.8	0.8
*SLA_max*	The maximum specific leaf area (m^2^ kg^−1^)	24	30	26
*SLA_min*	The minimum specific leaf area (m^2^ kg^−1^)	14	14	12
*AMAX*	The maximum assimilation rate (kg ha^−1^ h^−1^)	45	60	50
*R_max*	Maximum root depth (cm)	120	120	120

**Table 4 plants-12-00597-t004:** Statistical indices for simulated soil water content, nitrate concentration, leaf area index, yield, and crop N uptake under different cropping systems.

Items	*n*	*R* ^2^	Regression Equation	*nRMSE*(%)	*NSE*	*IA*
Soil water content (cm cm^−3^)	1100	0.519 **	y = 0.733x + 0.050	20.3	0.423	0.847
Soil nitrate concentration (mg kg^−1^)	44	0.317 **	y = 0.443x + 2.417	37.5	0.025	0.708
Above-ground Dry Mass (kg ha^−1^)	129	0.916 **	y = 0.906x + 1510	19.4	0.903	0.974
Leaf area index (cm^2^ cm^−2^)	117	0.726 **	y = 0.747x + 1.647	29.9	0.649	0.898
Yield (kg ha^−1^)	27	0.522 **	y = 0.838x + 854	12.9	0.433	0.844
Crop nitrogen uptake (kg ha^−1^)	77	0.828 **	y = 0.780x + 25.56	21.2	0.808	0.942

Note: **, significant at 0.01 probability level.

**Table 5 plants-12-00597-t005:** Water fluxes of the 120 cm soil profile simulated by the WHCNS model for five systems.

Systems	Growth Period	Crop	P	I	D	R	ET	Bal	Yield	WUE	PE
			mm						(kg ha^−1^)	(kg m^−3^)	
2H1Y_FP	2004.10–2005.6	WW	70	300	0	0	379	−8	6586	1.7	0.19
	2005.6–2005.10	SUM	247	132	0	0	327	52	8832	2.7	0.76
	2005.10–2006.6	WW	120	300	1	0	370	49	5610	1.5	0.32
	2006.6–2006.10	SUM	516	0	98	51	292	75	8609	2.9	1.76
	2006.10–2007.6	WW	130	300	47	11	392	−21	4675	1.2	0.33
	2007.6–2007.10	SUM	261	75	16	0	282	38	8596	3.0	0.93
	Summary		1344	1107	162	63	2042	185	42,908	2.1	0.66
2H1Y_WQ	2004.10–2005.6	WW	70	225	7	0	382	−94	6749	1.8	0.18
	2005.6–2005.10	SUM	247	132	1	0	322	56	9377	2.9	0.77
	2005.10–2006.6	WW	120	225	2.2	0	340	3	5996	1.8	0.35
	2006.6–2006.10	SUM	516	0	62	21	291	142	8833	3.0	1.77
	2006.10–2007.6	WW	130	225	26	12	389	−72	5872	1.5	0.33
	2007.6–2007.10	SUM	261	75	2	0	273	61	8244	3.0	0.96
	Summary		1344	882	100	33	1998	95	45,071	2.3	0.67
2H1Y_RI	2004.10–2005.6	WW	70	105	2	0	318	−144	6096	1.9	0.22
	2005.6–2005.10	SUM	247	132	1	0	312	67	9135	2.9	0.79
	2005.10–2006.6	WW	120	116	1	0	253	−18	4839	1.9	0.47
	2006.6–2006.10	SUM	516	0	48	20	289	159	8714	3.0	1.78
	2006.10–2007.6	WW	130	125	16	0	369	−130	5870	1.6	0.35
	2007.6–2007.10	SUM	261	60	2	0	257	62	7602	3.0	1.02
	Summary		1344	538	69	20	1798	−4	42,256	2.4	0.75
3H2Y	2004.10–2005.6	WW	70	105	0	0	342	−167	6026	1.8	0.21
	2005.6–2005.10	SUM	247	132	0	0	298	81	8879	3.0	0.83
	2006.4–2006.9	SPM	608	60	49	13	382	224	8059	2.1	1.59
	2006.10–2007.6	WW	130	125	22	0	377	−144	5872	1.6	0.34
	2007.6–2007.10	SUM	261	60	1	0	252	69	7407	2.9	1.04
	Summary		1317	482	7	13	1652	62	36,243	2.2	0.80
1H1Y	2005.4–2005.9	SPM	239	120	0	0	419	−60	7225	1.7	0.57
	2006.4–2006.9	SPM	608	0	54	11	368	176	7481	2.0	1.66
	2007.4–2007.9	SPM	258	60	0	0	369	−51	9040	2.4	0.70
	Summary		1105	180	54	11	1156	65	23,746	2.1	0.96

Note: SPM—spring maize; SUM—summer maize; WW—winter wheat; P—precipitation, mm; I—irrigation; D—drainage; R—runoff; ET—evapotranspiration; Bal = P + I – D – R − ET, mm; PE = P/ET; WUE = Yield/(ET × 10).

**Table 6 plants-12-00597-t006:** Nitrogen fluxes of the 120cm soil profile simulated by the WHCNS model for 5 systems.

Systems	Growth Period	Crop	Input (kg N ha^−1^)		Output (kg N ha^−1^)				N_bal_	NUE
			N_fer_	N_min_	N_den_	N_vol_	N_lea_	N_up_		(kg kg^−1^)
2H1Y_FP	2004.10–2005.6	WW	300	63	1	21	0	276	65	22.1
	2005.6–2005.10	SUM	250	112	4	26	0	225	107	34.6
	2005.10–2006.6	WW	300	84	6	19	2	283	74	18.1
	2006.6–2006.10	SUM	250	98	33	23	113	208	−29	22.8
	2006.10–2007.6	WW	300	97	15	20	54	326	−17	11.3
	2007.6–2007.10	SUM	250	87	21	19	18	217	62	31.2
	Summary		1650	540	80	127	187	1535	262	23.6
2H1Y_WQ	2004.10–2005.6	WW	202	65	3	17	2	267	−22	23.3
	2005.6–2005.10	SUM	237	141	4	29	1	222	122	36.7
	2005.10–2006.6	WW	194	107	4	13	3	249	33	22.4
	2006.6–2006.10	SUM	202	108	31	19	63	206	−9	27.7
	2006.10–2007.6	WW	206	94	14	11	29	291	−44	17.1
	2007.6–2007.10	SUM	202	106	8	21	3	216	61	33.4
	Summary		1243	621	63	110	100	1450	140	26.1
2H1Y_RI	2004.10–2005.6	WW	125	57	2	11	1	204	−36	28.0
	2005.6–2005.10	SUM	30	105	2	6	0	192	−64	45.8
	2005.10–2006.6	WW	105	78	1	7	0	188	−13	24.7
	2006.6–2006.10	SUM	110	160	26	12	13	205	16	34.2
	2006.10–2007.6	WW	170	109	10	10	5	274	−21	19.6
	2007.6–2007.10	SUM	40	117	4	7	1	177	−31	40.5
	Summary		580	626	45	52	20	1239	−149	31.2
3H2Y	2004.10–2005.6	WW	125	63	2	12	0	223	−48	25.5
	2005.6–2005.10	SUM	30	110	2	6	0	190	−57	45.0
	2006.4–2006.9	SPM	70	83	12	7	1	168	−35	43.0
	2006.10–2007.6	WW	170	69	7	10	1	243	−21	22.6
	2007.6–2007.10	SUM	40	126	5	7	0	164	−10	42.1
	Summary		435	451	28	41	1	987	−171	34.3
1H1Y	2005.4–2005.9	SPM	72	138	4	19	0	307	−121	21.8
	2006.4–2006.9	SPM	30	154	14	9	25	238	−101	26.2
	2007.4–2007.9	SPM	30	151	3	3	0	248	−73	35.5
	Summary		132	443	22	31	25	794	−296	27.3

Note: N_fer_—N fertilizer; N_min_—net mineralization; N_vol_—ammonia volatilization; N_den_—denitrification; N_lea_—N leaching; N_up_—crop N uptake; N_bal_—N balance; NUE—N use efficiency; NUE = Yield/(N_den_ + N_vol_ + N_lea_ + N_up_).

**Table 7 plants-12-00597-t007:** Integrated evaluation indexes for selecting scenarios simulated by the WHCNS model under different cropping systems.

Water Price	Systems	AF	EF	VCR	OOV	WOV
0.25 CNYm^−3^	2H1Y_FP	0.86	1.00	9.08	3.92	0.00
	2H1Y_WQ	0.92	0.69	12.51	3.07	1.00
	2H1Y_RI	0.94	0.29	23.07	1.49	3.00
	3H2Y	0.88	0.17	24.43	1.29	3.37
	1H1Y	0.64	0.19	45.49	0.00	5.35
0.5 CNYm^−3^	2H1Y_FP	0.86	1.00	7.14	3.76	0.00
	2H1Y_WQ	0.92	0.69	9.76	2.86	1.02
	2H1Y_RI	0.94	0.29	17.21	1.28	3.03
	3H2Y	0.88	0.17	17.69	1.14	3.35
	1H1Y	0.64	0.19	31.79	0.00	5.12
1 CNYm^−3^	2H1Y_FP	0.86	1.00	5.00	3.59	0.00
	2H1Y_WQ	0.92	0.69	6.77	2.66	1.05
	2H1Y_RI	0.94	0.29	11.42	1.06	3.05
	3H2Y	0.88	0.17	11.40	1.00	3.31
	1H1Y	0.64	0.19	19.84	0.00	4.89
1.5 CNYm^−3^	2H1Y_FP	0.86	1.00	3.85	3.51	0.00
	2H1Y_WQ	0.92	0.69	5.18	2.55	1.06
	2H1Y_RI	0.94	0.29	8.54	0.96	3.06
	3H2Y	0.88	0.17	8.41	0.93	3.29
	1H1Y	0.64	0.19	14.42	0.00	4.77
2 CNYm^−3^	2H1Y_FP	0.86	1.00	3.13	3.46	0.00
	2H1Y_WQ	0.92	0.69	4.20	2.49	1.07
	2H1Y_RI	0.94	0.29	6.82	0.90	3.06
	3H2Y	0.88	0.17	6.66	0.89	3.27
	1H1Y	0.64	0.19	11.32	0.00	4.70

Note: AF—agronomy effects; EF—environmental effects; VCR—value to cost ratio; OOV—the optimal osculating value; WOV—the worst osculating value. The values of AF, EF, and VCR were normalized.

## Data Availability

Not applicable.

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
