# Peer review of "Improvement of Water and Nitrogen Use Efficiencies by Alternative Cropping Systems Based on a Model Approach"

_plants, 2023, doi:10.3390/plants12030597_

Round 1

Reviewer 1 Report

Major Comments

1.    Although the authors mentioned adopting an experimental design when conducting the field experiment (Line 120), the statistical analysis was not covered in the Methodology section.

2.    The data presented in the Results Section was not statistically analyzed. Although the authors mentioned that one treatment was significantly higher and lower than the other treatments (e.g., Lines 422, 440, 481, 483), this was not supported by an appropriate statistical analysis.

3.    In most parameters, the authors predominantly presented the data by ranking the treatments rather than highlighting significant differences among treatments.

4.    The authors discussed the nitrogen fertiliser data (Lines 440–442), but the values were not reflected in Table 6.

Minor Comments

Abstract

1.    The precipitation range in 2H1Y (59–71%) was compared against average values in 3H2Y (76%) and 1H1Y (91%).

2.    The data presentation in Lines 24 -25 is unclear. The use of respectively in a series comparisons (more than three) could be confusing to the readers.

Introduction

1.    Add discussion about the use and importance of the WHCNS model in relation to this research work.

2.    Add calibration and validation of WHCNS as one of the objectives (Lines 82–83).

Methodology

1.    In Line 96, look for a typographical error: silt instead of slit.

2.    Two harvests in one year (lines 97-98) and winter wheat/summer maize (line 98) can be deleted as these were already mentioned earlier.

3.    It is possible to include information on how N was optimised (Line 112).

4.    It is randomised complete block design rather than a randomised block design.

5.    Spell out TDR (Line 134).

Results and Discussion

1.    Spell out LAI and put LAI in the parenthesis at Line 288.

2.    The data presentation in Lines 24–25 is unclear. The use of "respectively" in a series of comparisons (more than three) could be confusing to the readers.

3.    May check for grammatical (e.g., "N leaching losses were in line 55") and typographical errors (e.g., "line 596").

4.    Figure 1 may require you to spell out the M/YY label.

5.    The legend for 2004-05 wheat does not appear in the bar graphs, whereas the legend for 2007 maize contains two values.

6.    In Table 4, "above ground dry mass" should be written "above ground dry mass" to be consistent with other items (parameters).

7.    In Tables 5 and 6, 2H1Y_WQ should come after 2H1Y_FP to be consistent with Figure 1.

Author Response

Major Comments

  1. Although the authors mentioned adopting an experimental design when conducting the field experiment (Line 120), the statistical analysis was not covered in the Methodology section.

Response: Good suggestion. The description of statistical analysis was added in section 2.7

  1. The data presented in the Results Section was not statistically analyzed. Although the authors mentioned that one treatment was significantly higher and lower than the other treatments (e.g., Lines 422, 440, 481, 483), this was not supported by an appropriate statistical analysis.

Response: Good comment. We added the results of statistical analysis on crop yield and model performance evaluation in corresponding Figure 4 and Table 4, respectively. Considering the difference between the model method and field experiment, each output item of the model simulation for each treatment was only one value (it is no replication), so statistical analysis methods cannot be used for the results of the model output, and we checked and deleted the "significantly" in the sections of results and discussion.

Figure 4. Average annual crop yield under different cropping systems. Error bars represent the standard error. Different lowercase letters indicate significant differences among yields under different treatments (P < 0.05).

Table 4. Statistical indices for simulated soil water content, nitrate concentration, leaf area index, yield and crop N uptake under different cropping systems.

Items

n

R2

Regression equation

nRMSE(%)

NSE

IA

Soil water content (cm cm-3)

1100

0.519**

y=0.733x+0.050

20.3

0.423

0.847

Soil nitrate concentration(mg kg-1)

44

0.317**

y=0.443x+2.417

37.5

0.025

0.708

Aboveground dry mass(kg ha-1)

129

0.916**

y=0.906x+1510

19.4

0.903

0.974

Leaf area index (cm2 cm-2)

117

0.726**

y=0.747x+1.647

29.9

0.649

0.898

Yield (kg ha-1)

27

0.522**

y=0.838x+854

12.9

0.433

0.844

Crop nitrogen uptake (kg ha-1)

77

0.828**

y=0.780x+25.56

21.2

0.808

0.942

Note: **, significant at 0.01 probability level.

  1. In most parameters, the authors predominantly presented the data by ranking the treatments rather than highlighting significant differences among treatments.

Response:  Please see our response to question#2. The main reason is that the statistical analysis methods cannot be used for the results of model output.

  1. The authors discussed the nitrogen fertiliser data (Lines 440–442), but the values were not reflected in Table 6.

Response: We’d like to explain that the specific rate of N fertilizer application each time was also added in Figure 1.

Figure 1. Crop rotations and water and N inputs for five different cropping systems. Note: 2H1Y_FP, two harvests in one year of farmer’s practice; 2H1Y_WQ, two harvests in one year of Wuqiao system; 2H1Y_RI, two harvests in one year of reduced input; 3H2Y, three harvests in two years; 1H1Y, one harvest in one year; WW, winter wheat; SUM, summer maize; SPM, spring maize; M/yy, Month/Year.

Minor Comments

Abstract

  1. The precipitation range in 2H1Y (59–71%) was compared against average values in 3H2Y (76%) and 1H1Y (91%).

Response: There are three patterns in the 2H1Y system (FP: Farmer's Practice; RI: Reduced Input; WQ: Wuqiao system), both 3H2Y and 1H1Y systems had only one pattern, so we give the range for 2H1Y. In order to correspond to the average values of different cropping systems, we corrected the sentence as "However, local precipitation during the period of crop growth could only meet 65%, 76% and 91% of total water consumption for 2H1Y 3H2Y and 1H1Y systems, respectively".

  1. The data presentation in Lines 24-25 is unclear. The use of respectivelyin a series comparisons (more than three) could be confusing to the readers.

Response: Good suggestion. we deleted the data of the 1H1Y treatment.

Introduction

  1. Add discussion about the use and importance of the WHCNS model in relation to this research work.

Response: Good comment. The use and importance of the WHCNS model in relation to this research work has been added in the Introduction section as following:

Moreover, to explore the optimal cropping system with high yield and environmentally friendly, there is still a lack of quantitative analysis tools for soil water and N fluxes and their efficiencies for different cropping systems. The dynamic field observation of evapotranspiration (ET), water drainage, N losses, and crop biological indicators are time-consuming and costly. To overcome this shortcoming, the soil–crop system models were used to simulate crop growth and N losses in cropland under different water and N management and environmental conditions. Recently, an integrated soil-crop model (WHCNS, soil Water Heat Carbon Nitrogen Simulator) was developed based on the Chinese climate, soil types, and field management to optimize water and N management (Liang et al., 2016a). The model has been successfully applied to simulate water consumption, N loss, and water and N use efficiencies in different regions of China (Liang et al., 2019; Shi et al., 2020; Xu et al., 2020; Leghari et al., 2019; Meng et al., 2022).

  1. Add calibration and validation of WHCNS as one of the objectives (Lines 82–83).

Response: Agreed. We added it as one of the objectives of this study.

Methodology

  1. In Line 96, look for a typographical error: silt instead of slit.

Response: Followed.

  1. Two harvests in one year (lines 97-98) and winter wheat/summer maize (line 98) can be deleted as these were already mentioned earlier.

Response: Followed.

  1. It is possible to include information on how N was optimized (Line 112).

Response: Good comment. The optimal N fertilizer was based on the real-time monitoring of the soil mineral N content in the 0-90 cm soil profile. Crop N demand was determined by crop target yield and the amount of fertilizer application is the difference between crop N demand and soil N supply (Liu et al., 2008).

  1. It is randomised complete block design rather than a randomised block design.

Response: Agreed. We corrected it as "randomized complete block designs".

  1. Spell out TDR (Line 134).

Response: We added the whole name for TDR (time domain reflectometry).

Results and Discussion

  1. Spell out LAI and put LAI in the parenthesis at Line 375.

Response: Followed.

  1. The data presentation in Lines 24–25 is unclear. The use of "respectively" in a series of comparisons (more than three) could be confusing to the readers.

Response: Good suggestion. We deleted the data of the 1H1Y treatment.

  1. May check for grammatical (e.g., "N leaching losses were in line 455") and typographical errors (e.g., "line 596").

Response: Agreed. Both corrected.

  1. Figure 1 may require you to spell out the M/YY label.

Response: We have spelled out the "M/YY" as "Month/Year" and added it in the note.

  1. The legend for 06-07 wheat does not appear in the bar graphs in 3H2Y, whereas the legend for 2007 maize contains two values.

Response: We corrected them in Figure 4.

  1. In Table 4, "Aboveground Dry Mass" should be written "Above ground dry mass" to be consistent with other items (parameters).

Response: Agreed. Changed.

  1. In Tables 5 and 6, 2H1Y_WQ should come after 2H1Y_FP to be consistent with Figure 1.

Response: Agreed. Changed.

Reviewer 2 Report

This MS evaluated the yield, ET, WUE, N balance and NUE under three cropping system and different water and N application conditions using a calibrated crop water model, then optimized cropping system and water and nitrogen application scheduling were proposed by considering the trade-off in crop yield and environmental benefit. Results in this study could help farmers and managers to sufficiently use water and fertilizer under high yield production.

One main point for this MS is the wheat yield used for model calibration. The wheat yield ranged from 4600-6600 kg ha-1 in the study. To date, the yield has reached to approximately 7000-9000 kg ha-1. Given the N content in plant is the same, the total N uptake will be increased by approximately more than 50%, indicating a much higher N application at present. This should be discussed when using the recommended N input value. ET is not changed greatly based researches and my study. Yield for maize is ok though it is little lower than those at present. 

Other minor comments can be found in the pasted file.

Author Response

Comments and Suggestions for Authors

MAIN POINTS

  1. This MS evaluated the yield, ET, WUE, N balance and NUE under three cropping system and different water and N application conditions using a calibrated crop water model, then optimized cropping system and water and nitrogen application scheduling were proposed by considering the trade-off in crop yield and environmental benefit. Results in this study could help farmers and managers to sufficiently use water and fertilizer under high yield production.

Response: Thanks for your positive comments.

  1. The wheat yield ranged from 4600-6600 kg ha-1 in the study. To date, the yield has reached to approximately 7000-9000 kg ha-1. Given the N content in plant is the same, the total N uptake will be increased by approximately more than 50%, indicating a much higher N application at present. This should be discussed when using the recommended N input value. ET is not changed greatly based researches and my study. Yield for maize is ok though it is little lower than those at present.

Response: Good question. I agreed with your opinion. wheat yield in the 2000s was much lower than that in the 2020s mainly due to the different wheat varieties and field management. Many studies found that higher yield will result in higher crop N uptake. Considering the experiment was conducted many years ago, and the main objective of this study was to identify a sustainable cropping system by using the process-based model to compare crop yields, N losses, WUE, and NUE among different cropping systems and the findings have implications for today.

We also added the discussion on this question in the 4.4 section as following:

"However, considering the experiment was conducted many years ago, today the wheat yield reaches 8000-9000 kg ha-1, which is much higher than that of many years ago due to the different wheat varieties and field management practices. Many studies also found that higher yield will result in higher crop N uptake. The recommended system in this study should be tested based on the new wheat variety and field management conditions."

MINOR POINTS

Abstract

  1. Both crop system and irrigation-fertilizer practice should be presented.

Response: We corrected the sentence as "the 3H2Y cropping system with optimal irrigation and fertilization."

Introduction

  1. The logical relationship presented in the first sentence of the Introduction part should be corrected.

Response: Good suggestion. We restructured the Introduction part. Please see lines 39-41

  1. The three cropping systems should be defined when first used.

Response: Followed.

Methodology

  1. The three water and N input modes for the 2H1Y system should be described in detail, this information is very important for evaluating the experiment results and model results.

Response: Good comments. We added the description of water and N management information for the 2H1Y system, please see lines 188-191. The specific amounts and timing of irrigation and N fertilization each time were also added in Figure 1.

  1. Parameters' value for DL and Do should be cited.

Response: We added the related references.

Liang, H.; Hu, K.; Batchelor, W.D.; Qi, Z.; Li, B. An integrated soil-crop system model for water and nitrogen management in North China. Sci. Rep. 2016, 6, 25755.

  1. For the calculation of VCR, my suggestion is the total cost for each cropping system should be considered in the calculation, including harvest and sowing, labor, weed and pest control, as well as other field management cost.

Response: Good suggestion. Considering that the experiment was conducted many years ago and those key information was missing, so it is difficult to collect it again. We will pay more attention to the data collection in future studies.

Results and discussion

  1. The recommended NSE>36,and the NSE for nitrate concentration in this study(0.025) look small.

Response: Good question. In fact, the model performance criteria are different for different researchers. Yang et al. (2014) recommended the values of IA ≥ 0.75 and NSE ≥ 0 as the minimum threshold values for crop growth evaluation, while values of IA ≥ 0.60 and NSE ≥ -1.0 is the minimum threshold values for soil outputs evaluation. Kersebaum et al. (2007) compared the performance of 13 soil-crop models on soil water and N dynamics, and found that the negative value of NSE can be accepted. Therefore, considering the complexity of N transformation, the NSE values of nitrate concentration (0.025) in this study can be accepted.

Yang, J.M.; Yang, J.Y.; Liu, S.; Hoogenboom, G. An evaluation of the statistical methods for testing the performance of crop models with observed data. Agric. Syst. 2014. 127, 81–89.

Kersebaum, K.C.; Hecker, J-M.; Mirschel, W.; Wegehenkel, M. Modelling water and nutrient dynamics in soil–crop systems. Netherlands: Springer.2007.

  1. All discussion words in the result part should be moved into discussion section.

Response: Agreed. We have moved the discussion content to the Discussion section or deleted them.

  1. In table 7, The unit of water price should be Chinese yuan, and is better to converted to some international currency, for example US dollar.

Response: Agreed. We corrected “yuan” as “CNY”

Round 2

Reviewer 1 Report

I would like to thank the authors for appreciating and agreeing with all my comments. In its current form, I believe the manuscript has been sufficiently improved to warrant publication in Plants.